# APETx-Like Peptides from the Sea Anemone *Heteractis crispa*, Diverse in Their Effect on ASIC1a and ASIC3 Ion Channels

**DOI:** 10.3390/toxins12040266

**Published:** 2020-04-20

**Authors:** Rimma S. Kalina, Sergey G. Koshelev, Elena A. Zelepuga, Natalia Y. Kim, Sergey A. Kozlov, Emma P. Kozlovskaya, Margarita M. Monastyrnaya, Irina N. Gladkikh

**Affiliations:** 1G.B. Elyakov Pacific Institute of Bioorganic Chemistry, Far Eastern Branch of the Russian Academy of Science, 690022 Vladivostok, Russia; zel@piboc.dvo.ru (E.A.Z.); natalya_kim@mail.ru (N.Y.K.); kozempa@mail.ru (E.P.K.); rita1950@mail.ru (M.M.M.); 2Shemyakin-Ovchinnikov Institute of Bioorganic Chemistry, Russian Academy of Science, 117997 Moscow, Russia; sknew@yandex.ru (S.G.K.); serg@ibch.ru (S.A.K.)

**Keywords:** sea anemone, APETx-like toxins, acid-sensing ion channels, electrophysiology, molecular modeling

## Abstract

Currently, five peptide modulators of acid-sensing ion channels (ASICs) attributed to structural class 1b of sea anemone toxins have been described. The APETx2 toxin is the first and most potent ASIC3 inhibitor, so its homologs from sea anemones are known as the APETx-like peptides. We have discovered that two APETx-like peptides from the sea anemone *Heteractis crispa*, Hcr 1b-3 and Hcr 1b-4, demonstrate different effects on rASIC1a and rASIC3 currents. While Hcr 1b-3 inhibits both investigated ASIC subtypes with IC_50_ 4.95 ± 0.19 μM for rASIC1a and 17 ± 5.8 μM for rASIC3, Hcr 1b-4 has been found to be the first potentiator of ASIC3, simultaneously inhibiting rASIC1a at similar concentrations: EC_50_ 1.53 ± 0.07 μM and IC_50_ 1.25 ± 0.04 μM. The closest homologs, APETx2, Hcr 1b-1, and Hcr 1b-2, previously demonstrated the ability to inhibit hASIC3 with IC_50_ 63 nM, 5.5, and 15.9 μM, respectively, while Hcr 1b-2 also inhibited rASIC1a with IC_50_ 4.8 ± 0.3 μM. Computer modeling allowed us to describe the peculiarities of Hcr 1b-2 and Hcr 1b-4 interfaces with the rASIC1a channel and the stabilization of the expanded acidic pocket resulting from peptides binding which traps the rASIC1a channel in the closed state.

## 1. Introduction

Acid-sensing ion channels (ASICs) activated by extracellular pH decrease are members of the amiloride-sensitive degenerin/epithelial sodium channels (DEG/ENaC) superfamily. Nine ASICs isoforms (1a,b, 2a,b, 3a–c, 4, 5) encoded by five genes (ACCN1-5) have been detected in mammals [1,2]. Among all known isoforms, ASIC1a and ASIC3 are key isoforms expressed in the neurons of both the central (CNS) and peripheral nervous systems (PNS) [3]. Homomeric ASIC1a and ASIC3 channels are the most rapidly activated/inactivated channels and are sensitive to the slightest deviation of the pH value from the physiological condition, pH_50_ 5.8 and 6.4, respectively [4].

Acidification of the synaptic cleft during synaptic transmission has proved to be sufficient for ASICs activation and induction of the postsynaptic current [4,5]. It supports the ASICs contribution to synaptic neurotransmission and plasticity (underlying memory and learning), fear, anxiety, and addiction-related behavior [6,7]. Homomeric ASIC1a and heteromeric ASIC1a/2a are the major channels of the CNS [8]. Neuron damage was revealed to be associated with the activation of ASIC1a due to the prolonged acidosis resulting from pathological conditions (ischemic stroke, epilepsy, multiple sclerosis, Parkinson’s disease, etc.) [4,5,9]. This makes ASIC1a a promising target for neuroprotective interventions. Homomeric ASIC3 and heteromeric ASIC3-containing channels of the PNS are involved in the perception of pain associated with tissue acidosis caused by mechanical damage, inflammation, tumor, etc. [6]. It has been determined that the peculiarity of ASIC3 channels is their ability to generate sustained currents in addition to the conventional transient currents. This phenomenon defines the functional consequences of ASIC3 activation [3,4].

Sixteen crystal structures of closed, desensitized and opened [10] states of cASIC1 channel, including complexes with proteinaceous (PcTx1 from *Psalmopoeus cambridgei* [11], MitTxα/β from *Micrurus tener tener* [12], mambalgin-1 from *Dendroaspis polylepis* [13]) and nonproteinaceous ligands (amiloride [12]) have been determined to date. It has been reported that functional ASIC channels are formed by three subunits. The individual subunit resembles a hand that clutches a ball: the extracellular region contains the so-called wrist, palm, finger, knuckle, thumb, β-ball domains, and the transmembrane domain includes two transmembrane helices with short intracellular termini [14]. The thumb, β-ball, finger domains of one subunit, and the upper palm domain of the neighboring subunit form an acidic pocket that accumulates acidic residues and serves as a potential pH sensor [14,15].

Animal toxins produced by spiders, snakes, and sea anemones are considered molecular instruments for the study of ASICs gating and promising drug precursors [5,16,17]. To date, the Hi1a and PcTx1 toxins have proved to be effective neuroprotectors in models of cerebral ischemia, retinal stroke, optic nerve crush, and Parkinson’s disease [16,18], while APETx2 (*Anthopleura elegantissima*) and mambalgins are considered to be prospective analgesic compounds [16,19]. Peptides from spiders and snakes inhibit mostly mammalian ASIC1a channels [20]. Two of ASIC1a inhibitors from spiders, PcTx1 and Hm3a (*Heteroscodra maculata*), promote desensitization of ASIC1a [21]. The spider toxin Hi1a (*Hadronyche infensa*) and snake toxin mambalgin-1 stabilize the closed channels state [13,20,22]. In contrast, the heterodimeric MitTxα/β complex from the Texas coral snake activates ASIC1a nine times more powerful than ASIC3 [20].

For a long time, the sea anemone toxins, APETx2, Hcr 1b-1 (*Heteractis crispa*), and Ugr 9a-1 (*Urticina grebelnyi*), have been known as selective ASIC3 inhibitors without an ASIC1a effect [19,23,24]. Later, we have shown that the APETx-like peptides from sea anemone *H. crispa*, Hcr 1b-2, Hcr 1b-3 and Hcr 1b-4, are capable of inhibiting ASIC1a, and Hcr 1b-2 inhibits the ASIC1a subtype more prominently than ASIC3. In addition, Hcr 1b-2 has demonstrated an anti-hyperalgesic effect, significantly reducing the pain threshold for animals in the acid-induced muscle pain model [25].

In this work, we have investigated in depth the electrophysiological effects of two previously isolated peptides, Hcr 1b-3 and Hcr 1b-4, on homomeric rASIC1a and rASIC3 channels expressed in *Xenopus laevis* oocytes and found a fundamental distinction between their action modes. Additionally, we have constructed theoretical models of peptides Hcr 1b-2 and Hcr 1b-4 complexes with the rASIC1a channel that provide insights into the molecular mechanism of the ASICs modulation by *H. crispa* peptides.

## 2. Results

### 2.1. Structural and Computational Analyses of APETx-Like H. crispa Toxins

Among the great diversity of sea anemone peptide structures, the *H. crispa* toxins (HcrTxs) affecting ASIC channels should be attributed to the structural class 1b [26,27]. HcrTxs share the best identity with the sea anemone toxins of class 1b, such as antimicrobial peptide crassicorin-I from *Urticina crassicornis* [28], ion channel modulators APETx1–APETx4 from *A. elegantissima* affecting ASICs and/or voltage-gated potassium and/or sodium channels [29,30,31,32,33], and several functionally uncharacterized APETx-like peptides identified in transcriptomes of sea anemones *A. elegantissima*, *Bunodosoma granuliferum* [34], *Anemonia sulcata* [35], and in the proteome of *Bunodosoma cangicum* [36]. The HcrTxs structural identity with peptides from other species reaches 46%–56%, while HcrTxs identity between themselves is more prominent and amounts to 78%–98%. Thus, the Hcr 1b-2 peptide is distinguished from Hcr 1b-3 by the Asn22Asp point mutation and from Hcr 1b-4 by nine amino acid substitutions. The most important is a homology to the well-studied APETx2 toxin that affects the same molecular target — ASICs [23,32]. Therefore, it was essential to find common features between HcrTxs and APETx2 that may underlie their inhibitory effect on ASICs. The sequence alignment (Figure 1) shows moderated structural homology of primary structures in all Cys residues and in the majority of Gly and aromatic residues. Despite this fact, Arg17, shown to be crucial for the APETx2 inhibitory activity toward rASIC3 [31], was substituted for Leu in HcrTxs. Therefore, it would be reasonable to generate 3D structures of HcrTxs and compare them with the APETx2 structure.

It is known that sea anemone toxins of structural class 1b, APETx1, APETx2 or BDS-I from *A. sulcata* [31,38,39,40], are β-structured. Analysis of secondary structure using circular dichroism (CD) spectroscopy revealed that secondary structure contents of HcrTxs is typical for peptides of structural class 1b [33]. Obtained CD spectra of HcrTxs were characterized by a large negative ellipticity in the vicinity of 205 nm and positive ellipticity around 230 and 190 nm (Figure 2). In general, they indicate predominant content of the β-strands and β-turns (about 40% and 20%, respectively) (Table 1) changing the direction of the peptide backbone. Less than 8% of the HcrTxs peptide chain adopts α-helical conformation, which is not observed in the structure of APETx1, APETx2 or BDS-I.

Protein Data Bank (PDB) contains 3D structures of three molecules sharing 54%–51%, 49%–46%, and 39% of sequence identity with HcrTxs: APETx1 (PDB ID 1WQK) [39], APETx2 (1WXN and 2MUB) [31,38], and BDS-I (1BDS and 2BDS) [40]. 3D models of Hcr 1b-2 and Hcr 1b-4 were based on the 3D structure of APETx2 (1WXN) [38]. The model of Hcr 1b-3 was generated from Hcr 1b-2 by the Asn22Asp substitution. The template model was chosen based on the minimal root mean square deviation (RMSD) value and backbone conformations occupying “allowed” regions of the Ramachandran plot. Homology models derived from the 3D structure of APETx2 seem to be the most reliable, which is consistent with the common molecular target of APETx2 and HcrTxs. They had no conformational hindrances and were in acceptable agreement with the CD spectroscopy data (Table 1).

Similarly to other APETx-like peptides, HcrTxs form four-stranded antiparallel β-sheets cross-linked by three disulfide bridges (C1–C5, C2–C4, C3–C6) [31,38,39,40]. Conforming to the obtained models, the short α-helix (22-24 aa) of HcrTxs localized within the flexible region between the second and third strands (Figure 3a). Hcr 1b-2, -3, -4, and APETx2 are very basic peptides (theoretical pI being of 9.53, 9.23, 9.26, and 9.33, respectively) [25]. Basic-aromatic clusters play an important role in the formation of complexes between the ASIC channel and peptide toxins PcTx1, mambalgin-1, Ugr 9a-1, and APETx2 [13,20,31]. The putative site of APETx2 interaction with the ASIC3 channel identified by scanning mutagenesis consists of Thr2, Phe15-Tyr16-Arg17, Phe33, and Ley34 [31] (Figure 3b).

There is an analogous hydrophobic patch on the surfaces of Hcr 1b-2, -3, and -4 molecules including Phe15-Met16-Leu17, Phe33-Leu34 (Hcr 1b-2 and -3) or Phe15-Met16-Leu17, Phe33-Met34 (Hcr 1b-4) (Figure 3b). Although Hcr 1b-2 and Hcr 1b-3 are distinguished by Asn22 substitution to Asp only, one can see a difference in their molecule surface electrostatic and hydrophobic properties distribution. Analysis of peptides’ intramolecular contacts showed the rearrangement of the interactions network. In particular, the Hcr 1b-3 residues Lys40 and Asp22 were found to connect indirectly via the Tyr26 benzene ring through pi-anion and pi-alkyl interactions, respectively, as opposed to Tyr26 of Hcr 1b-2 involved in the pi-pi stacked interaction with Tyr24 which, in turn, is linked up to β-sheet by pi-alkyl interaction with Val12. However, compared to APETx2, the spatial models of HcrTxs demonstrate distinct localization of charged residues. There are no positively charged residues near this hydrophobic patch in the structures of Hcr 1b-2 or Hcr 1b-3. Four basic residues of Hcr 1b-2 or Hcr 1b-3, Lys 5, His7 (with a weakly ionized imidazole group), and Lys40-Lys41 are placed on the opposite side of the molecule (relative to Arg17 of APETx2) in close proximity to Cys6, Ile10, Val12, Leu28, and Cys30 (Figure 3b). Two clusters of basic residues are localized on the opposite faces of Hcr 1b-4 with two positively charged C-terminal residues similar to those of HcrTxs and Arg19, and with His31 adjacent to Phe15-Met16-Leu17, Cys20, Ile36, and Phe33-Met34 (Figure 3b). Regarding this observation, the directions of the dipole moments of Hcr 1b-2 (or Hcr 1b-3), Hcr 1b-4 and APETx2 are distinct (Figure 3a).

### 2.2. Electrophysiological Effects of HcrTxs on ASIC1a and ASIC3 Channels

The study of the pharmacological profile of the Hcr 1b-3 and Hcr 1b-4 peptides from *H. crispa* was carried out on two major ion channel subtypes, ASIC1a and ASIC3. Homomeric ASIC1a channels expressed in *X. laevis* oocytes were activated by a rapid pH drop from 7.8 to 5.5 in 15 s after peptide application. To analyze the effect of HcrTxs on the transient component of the ASIC3 current, the channels were activated by rapid extracellular acidification from pH 7.8 to 4.0, but effects on the sustained ASIC3 current were not studied. The inhibitory activity of three class 1b sea anemone toxins, APETx2, Hcr 1b-1, and Hcr 1b-2, towards ASICs was previously assayed in electrophysiological experiments (Table 2). In addition, we previously reported that the Hcr 1b-3 and Hcr 1b-4 peptides inhibited about 54% and 80% of the ASIC1a current, respectively [25].

The next in-depth electrophysiological investigations show that Hcr 1b-3 inhibits the ASIC1a current in a concentration-dependent manner with IC_50_ of 4.95 ± 0.19 μM and nH of 1.20 ± 0.05. The ASIC1a current inhibition by the toxin in concentrations from 0.1 to 120 μM was not complete, and maximal inhibitory effect reached 70% at the concentration of 120 μM (Figure 4a,e, Table 2). The inhibition of ASIC1a was completely reversible. The amplitude of the recorded current before and after the application of Hcr 1b-3 was nearly identical.

Hcr 1b-4 proved to inhibit the ASIC1a current in a manner similar to Hcr 1b-3 (Figure 4b). The effect was reversible, and the concentration–response relationship estimates that a maximal inhibition was 86% with IC_50_ value 1.25 ± 0.04 μM and nH 1.08 ± 0.03 (Figure 4e). It can be seen that the efficacy of the ASIC1a inhibition by Hcr 1b-4 is higher than by Hcr 1b-3 within all applied toxin concentrations (Table 2).

The effect of Hcr 1b-3 and Hcr 1b-4 on the transient component of ASIC3 current was measured as well. Electrophysiological experiments demonstrated that Hcr 1b-3 reversibly inhibited the transient current through ASIC3 channels (Figure 4c). The concentration–response relationship gives IC_50_ 17 ± 5.8 μM (Figure 4e). The maximal inhibition of the ASIC3 transient current by 120 μM of Hcr 1b-3 was about 80%. The efficiency of the inhibition of the ASIC1a and ASIC3 transient current by Hcr 1b-3 was comparable at minimal and maximal concentrations, but intermediate concentrations of Hcr 1b-3 inhibited ASIC1a more effectively than ASIC3.

The most interesting result was obtained during the Hcr 1b-4 effect elucidation on ASIC3. We postulated this molecule as the first peptide toxin potentiating the proton-evoked ASIC3 current. It did not activate the ASIC3 channel itself when applied at pH 7.8 but produced a remarkable potentiation of the transient current resulting from the acidic pulse (pH 4.0) (Figure 4b). At the maximal applied concentration, 120 μM, Hcr 1b-4 elicited responses that were twice as high as those produced by extracellular protons. The concentration–response data of the Hcr 1b-4 effect on ASIC3 channels was well-fitted with the curves, and the following parameters were calculated: EC_50_ 1.53 ± 0.07 μM and nH 1.30 ± 0.06 (Figure 4f). Thus, in contrast to Hcr 1b-1, Hcr 1b-2, and Hcr 1b-3, the Hcr 1b-4 peptide demonstrated an opposite effect on ASIC1a and ASIC3. Among all HcrTxs, it showed lower concentration of 50% effects on ASICs.

### 2.3. Molecular Modeling of the HcrTxs Interaction with the ASIC1a Channel

The details of the animal toxins’ interaction with ASIC channels were investigated by the X-ray crystallographic analysis or cryo-electron microscopy of the toxin complex with chicken ASIC1 (cASIC1) [11,12,13,43]. Since the 3D structure of other ASICs subtypes including ASIC3 has not yet been determined, we generated and analyzed theoretical models of Hcr 1b-2 and Hcr 1b-4 complexes with rASIC1a. The Hcr 1b-2 and Hcr 1b-4 peptides were selected for in silico analysis as the two most potent ASIC1a inhibitors (given that Hcr 1b-2 and Hcr 1b-3 are equipotent) with the remarkable difference in surface electrostatic and hydrophobic properties (Figure 3b). Additionally, the effects of Hcr 1b-2 and Hcr 1b-4 on ASIC3 current are distinct, and we believe that there is some correlation between the molecular basis of the ASIC1a inhibition and ASIC3 modulation by Hcr 1b-4.

According to the conditions of our electrophysiological experiments, the cASIC1a structure in the closed conformation (PDB ID 6AVE and 5WKV) [15], which is typical at high pH values, was chosen as the template for the generation of the rASIC1a homology model. The amino acid sequences of rASIC1a and cASIC1 were aligned using the BLAST algorithm [44] sequence identity of template and rASIC1a was 90%. The RMSD value for 175 cα atoms of the rASIC1a model relative to the prototype was 2.14 Å. The conformational analysis reveals a sufficient rASIC1a model quality with 94.2% of the residues having occupied the most favored regions and 5.8% fell into allowed areas of the Ramachandran plot, as well as no steric hindrance. The rigid body molecular docking demonstrated that the Hcr 1b-2 or Hcr 1b-4 molecule could adopt two possible sites of the rASIC1a homotrimer localized in the area of the thumb domain and within the acidic pocket of the channel (Appendix A). One of the predicted binding sites of Hcr 1b-2 and Hcr 1b-4 partially overlapped with that of PcTx1 [11], and the other overlapped with the interface of cASIC1 and mambalgin-1 [13] or MitTxα/β [12].

The evaluation of the physicochemical properties of snake and spider toxins affecting ASIC1a was carried out to identify the possible regions with similar amino acids. The electrostatic interaction between basic toxins and negatively charged ASICs is believed to be a main driving force of the channel recognition by animal toxins [20]. It should be noted that the dyads of positively charged residues on C-termini of Hcr 1b-2 or Hcr 1b-4 molecules could mimic Arg26-Arg-27-Arg28 residues on the top of the PcTx1 β-hairpin loop reaching the acidic pocket depth in the ASIC channel [11] (Figure 3a and Figure 5). In addition, the basic residues spaced 20–25 Å apart from each other, to mirror the binding pattern of the snake toxins with the thumb domain, were not found on the surfaces of compact sea anemone peptides. On the other hand, the intermolecular interactions of Hcr 1b-4 in these docking-predicted solution clusters (Appendix A) contributed from −86.35 to −50.7 kcal/mol at the area of the channel acidic pocket as well as −42.30 to –22.63 kcal/mol at another site while the Hcr 1b-2 contribution was estimated as –47.26 to –23.15 kcal/mol and –12.63 to –9.19 kcal/mol, respectively. Therefore, we conclude that Hcr 1b-2 and Hcr 1b-4 most likely insert themselves between two subunits within the acidic pocket, similar to PcTx1.

The analysis of the architecture of the rASIC1a–Hcr 1b-2 and rASIC1a–Hcr 1b-4 complexes at pH 7.8 and 5.5 derived from docking and refined with molecular dynamics (MD) simulations has revealed that the peptides form multiple contacts with the interface of rASIC1a subunits (Appendix A). Among the six amino acid residues of Hcr 1b-2 contributing to complex formation, all basic residues (Lys40-Lys41, Lys5) as well as Tyr9 and His7 proved to be hot spots (Figure 6a, Appendix A). The positively charged (Arg41, Arg19, Lys40) and aromatic residues (Tyr9, Tyr7), as well as the regions Ser22-Gly23-Tyr24 and Asn27-Ley28, shape the interface of Hcr 1b-4 and rASIC1a. These residues seem to be the key for the Hcr 1b-4 interaction with rASIC1a (Figure 6b, Appendix A). It should be noted that most of them are variable between Hcr 1b-2 and Hcr 1b-4 (Figure 5). Ile10 and Lys41 of Hcr 1b-2 as well as Lys40, Arg41, and Tyr9 of Hcr 1b-4 make contact with Asp237 of the acidic loop and Asp349 or Asp345 of the thumb domain of the rASIC1a channel (Figure 6a,b, Appendix A). These acidic residues are known as proton sensors, and Asp237–Asp349 as well as Glu238–Asp345 proton-mediated carboxyl-carboxylate pairings stabilize the collapsed conformation of the pocket [15,17].

It has been demonstrated that PcTx1 interaction with the proton sensors of rASIC1a locks proton-mediated interactions between the thumb and finger domains in the course of acidification [46,47]. Another similarity between the Hcr 1b-4 and PcTx1 interactions with the ASIC1a channel is the involvement of His173 (Appendix A) in the complex formation [46]. Since this residue is present only in mammalian ASIC1a, it seems to be important for specific modulation of homomeric ASIC1a channels.

## 3. Discussion

Currently, seven ASICs modulators produced by the sea anemones have been discovered [20,24,25]. Peptides Hcr 1b-3 and Hcr 1b-4, electrophysiologically examined in this study, are the closest homologues of Hcr 1b-1 and Hcr 1b-2 (78%–97% of identity) as well as APETx2 (46%–49% of identity) that are attributed to β-defensin-like peptides. APETx2 is the first and the most potent inhibitor of the ASIC3 current transient component with no effect on the ASIC3 sustained component or the ASIC1a current [20] (Table 2). Despite the absence of the crucial basic residue Arg17 [31] in the sequences of HcrTxs (Figure 1), our results demonstrate that HcrTxs retain the ability to modulate ASIC channels, even though Hcr 1b-1 and Hcr 1b-2 are 31- and 252-times, respectively, less potent inhibitors of transient ASIC3 currents compared to APETx2 [20,24,25] (Table 2).

In previous work we have shown that sea anemone peptides also inhibit homomeric rASIC1a channels [25]. Hcr 1b-3 and -4 affect rASIC1a channels in a similar manner as Hcr 1b-2 does [25]. The most complete inhibition of the rASIC1a current has been achieved by using Hcr 1b-4 in all assayed concentrations. The Hcr 1b-2 and -3 efficacy is nearly equal. The potency of Hcr 1b-2, -3, and -4 follows the same pattern. The Hcr 1b-4 peptide is a four-times more active inhibitor of rASIC1a than Hcr 1b-2 and Hcr 1b-3, whose IC_50_ values are identical (Table 2). The negligible difference between the Hcr 1b-2 and -3 effects is not surprising, taking into account that these two toxins differ by a single substitution (Figure 1). The multiple natural point substitutions of Hcr 1b-4 slightly affect both potency and efficacy of the rASIC1a inhibition (Table 2). Despite the discovered highest potency of Hcr 1b-4, rASIC1a modulators from the sea anemones have proved to remain much less active than various spider or snake toxins, whose IC_50_ values vary from 0.4 to 55 nM [20].

The effect of Hcr 1b-3 and -4 on the rASIC3 current is opposite. The Hcr 1b-3 peptide inhibiting the transient rASIC3 current with IC_50_ 17 ± 5.8 μM is the least potent ASICs inhibitor among sea anemone peptides. Its activity is inferior not only to Ugr 9a-1 (IC_50_ 10 μM) [48] or APETx2 [32], but also to Hcr 1b-1 and Hcr 1b-2 [24,25] (Table 2).

Surprisingly, Hcr 1b-4 effectively enhances the acid-evoked current through homomeric rASIC3 channels. It is known that the sea anemone toxins inhibiting ASIC3 are able to potentiate ASIC1b and ASIC2a currents (APETx2) [23] whereas ASIC1a inhibitors from spiders potentiate ASIC1b (PcTx1, Hi1a, Hm3a) and heteromeric ASIC1a/1b (Hm3a) channels. It is noteworthy that to increase the current through the ASICs it takes an order of magnitude higher concentration of toxins than to inhibit it [20]. In contrast, Hcr 1b-4 is an equally potent modulator of ASIC1a and ASIC3 subtypes (Table 2). No ASIC3 potentiator from the sea anemones, snakes or spiders has been previously described still. The single known ASICs activator, heteromeric complex MitTxα/β, was reported to elicit currents itself [49]. Hcr 1b-4 does not demonstrate such a channel-agonistic effect at neutral pH values. Therefore, peptide Hcr 1b-4 has a unique pharmacological profile as an inhibitor of ASIC1a and potentiator of ASIC3. The EC_50_ value of MitTxα/β for activation of ASIC3 is maximal (830 ± 250 nM) compared with that of the other ASICs subtypes [49], however, it is lower than the EC_50_ of Hcr 1b-4 (Table 2). At the same time, a proteinaceous ASIC3 potentiator, neuropeptide RPRFa from *Conus textile*, has been recently identified (EC_50_ 3–4 μM) [50]. Its effect on the proton-evoked ASIC3 current is similar to that of Hcr 1b-4, however, it is three-fold less active towards ASIC3 than Hcr 1b-4. Both MitTxα/β and RPRFa have demonstrated pain enhancement [49,50]. Hcr 1b-4 might have a different biological effect regarding its opposite action on ASIC1a and ASIC3 currents within the same concentration range (Table 2).

As we previously mentioned, HcrTxs are weak ASICs modulators demonstrating their effects at micromolar concentration. At the same time, APETx-like peptides were shown to modulate several ion channels, ASICs, Na_V_, and K_V_. Toxin APETx2 inhibits ASIC3 (IC_50_ 63 nM [32]), as well as voltage-gated sodium channels Na_V_1.2 (IC_50_ 114 ± 25 nM) and Na_V_1.8 (IC_50_ 55 ± 10 nM) [30] at nanomolar concentrations, but higher concentrations of APETx2 inhibit voltage-gated potassium channel hERG (IC_50_ 1.21 ± 0.05 μM) [31]. Unfortunately, it was not reported if other APETx-like peptides affect ASICs. Highly homologous peptides APETx1, APETx3 and APETx4 primarily inhibit K_V_11 (IC_50_ 34 nM for hERG [33]), potentiate TTX-sensitive Na_V_ (EC_50_ 2265 ± 185 nM for Na_V_1.6 [30]), and inhibit K_V_10.1 (IC_50_ 1.1 μM [29]), respectively, but in general each of these peptides modulate several subtypes of Na_V_ and K_V_ channels. Finally, toxin BDS-I initially characterized as a K_V_3 channel blocker (IC_50_ values 47 nM [51] and 220 nM [52] for K_V_3.1 and K_V_3.4, respectively) also potentiates TTX-sensitive Na_V_ channels (EC_50_ about 3 nM for Na_V_1.7) and weakly inhibits TTX-resistant Na_V_ [53]. Collectively, these data provide evidence that HcrTxs may modulate ion channels other than ASICs at a lower concentration range. It is noteworthy that ion channels targeted by APETx-like peptides belong to two different superfamilies: voltage-gated ion channels (Na_V_ and K_V_) and DEG/ENaC (ASICs). As far as we know, there is no data describing modulation of any member of the DEG/ENaC superfamily, except for ASICs by APETx-like peptides.

At present, data concerning interactions of the sea anemone toxins and ASIC channels is limited, and scanning mutagenesis of APETx2 revealed a cluster of aromatic and basic residues that mediate APETx2 interaction with ASIC3 [31]. It should be noted that the structures of ASIC channel complexes with the sea anemone toxins are much less studied compared to those of spider or snake peptides.

To explain the inhibitory effect of HcrTxs on rASIC1a channels, we generated theoretical models of the Hcr 1b-2 and Hcr 1b-4 complexes with this channel using homology modeling, molecular docking, and MD simulation approaches. Both sea anemone peptides were found to be localized between two adjacent channel subunits forming multiple contacts with the rASIC1a thumb, finger and palm domains (Figure 6a,b). The relative position and conformation of thumb, finger, β-ball, and palm domains after toxin binding is believed to underlie the complex pharmacology of PcTx1, increasing the ASICs apparent affinity for protons and stabilizing an open or desensitized state of the channel [11,20]. The positively charged C-termini of the Hcr 1b-2 or Hcr 1b-4 molecules, attracted by oppositely charged residues of the acidic loop and α5 helix (Glu177, Asp237 or Glu235, Asp237, Asp349, respectively), enter shallowly into the acidic pocket and form ionic interactions and H-bond networks. This makes peptides easier to permeate between the palm domain and α5 helix. The finger domain is also anchored by His7 of Hcr 1b-2 or Leu28 of Hcr 1b-4 (Appendix A).

It is noteworthy that the majority of the residues involved in hotspot interactions of Hcr 1b-2 and Hcr 1b-4 with rASIC1a are variable (Figure 5). They are localized on the opposite side of molecules as compared with the functionally important basic-aromatic cluster of APETx2 [31] (Figure 3b). This is due to the distinct electrostatic properties of the Hcr 1b-2, Hcr 1b-4 and APETx2 molecules. An electrophysiological estimation of the HcrTxs’ action on rASIC1a showed a higher potency of Hcr 1b-4 (Figure 4e). This correlates well with the simulation results that demonstrated that the significantly more potent Hcr 1b-4 modulator operates on rASIC1a through a wider network of intermolecular interactions with a higher energy contribution than those of Hcr 1b-2 (Figure 6a,b, Appendix A). Interestingly, the Hcr 1b-4 dipole moment in a complex with rASIC1a is directed (in contrast to that of the Hcr 1b-2) towards the center of the channel pore, which apparently provides evidence of the crucial role of the HcrTxs molecular electrostatic potential distribution for their channel activity.

The conformation of the cavity formed by the thumb, finger, and β-ball domains of one subunit and the palm region of the neighboring subunit determines the state of the channel gate at various pH values. The expansion of the channel fenestrations and pores is the result of the wave-like spread of motions transmitted from the extracellular to the transmembrane domain [15,17]. The collapse of the acid pocket that occurred in response to acidosis initiates a conformational ASICs channel transition, leading to channel opening or desensitization [15].

According to our theoretical results, the Hcr 1b-2 and Hcr 1b-4 peptides stabilize the non-conducting state of the rASIC1a channel with the expanded acidic pocket, even at a pH value reduced to 5.5. Despite the observed difference in Hcr 1b-2 and Hcr 1b-4 molecule poses, rASIC1a channel spatial structures (except for the acidic pocket) in complexes with both peptides changed slightly at high and low pH values. In the obtained models, the channel pore is restricted by the closed gate between the extracellular and intracellular vestibules. The pore profiles of the resting and desensitized channels are similar and have a minimum distance of 6.2–6.4 Å between the three Asp433 residues of cASIC1a [15,54]. Similar values, 5.4–6.9 Å (Asp432 of rASIC1a), were detected in rASIC1a–Hcr 1b-2 and rASIC1a–Hcr 1b-4 complexes at pH 7.8 and 5.5. The distance separating these residues in the open cASIC1 channel was 12.3–13.8 Å [11,12], twice as much as we observed in complexes with Hcr 1b-2 and Hcr 1b-4. The closed gate of the rASIC1a–Hcr 1b-2 and rASIC1a–Hcr 1b-4 complexes is stabilized by a circle pattern of inter- and intra-subunit hydrogen bonds and ionic interactions between residues Glu277‒Arg279, Gln278‒Arg369‒Glu79, and Glu416‒ Gln278 within the central vestibule of the rASIC1a channel, which hinders the opening of the pore (Figure 6d).

The main feature of rASIC1a complexes with HcrTxs is the remarkably expanded acidic pocket (Figure 6c) in comparison with the open channel conformation. The analogous conformation of the thumb α5 helix is elicited by the mambalgin-1 binding that stabilizes the closed state of the channel. Meanwhile, the localization of HcrTxs and snake toxin binding site on ASIC1 differs significantly [13]. The distance between the C alpha atoms of Asp237 and Asp345 or Asp349 (proton sensors) in the complex with Hcr 1b-2 and Hcr 1b-4 is 11–12 Å versus 7.2–9.5 Å in 3D structures of the open or desensitized cASIC1 channel.

We suppose that Hcr 1b-2 and Hcr 1b-4 binding hinders the collapse of the rASIC1a acidic pocket upon the exposure of protons. The Hcr 1b-2 and Hcr 1b-4 molecules trap the thumb α5 helix in the intermediate position between the expanded pocket of the resting channel and the collapsed pocket of the open channel [11,12,15] which makes the rASIC1a channel unable to produce an extracellular acidification response.

## 4. Conclusions

To date, there are five toxins affecting ASIC channels in the structural class 1b. Electrophysiological investigation of two of these peptides from *H. crispa*, Hcr 1b-3 and Hcr 1b-4, showed their ability to inhibit homomeric ASIC1a and modulate ASIC3 channels at micromolar concentrations. This was the first time showing the Hcr 1b-4 peptide being able to both inhibit rASIC1a and potentiate rASIC3 channels. The obtained results of electrophysiological experiments complete the previously reported data on the inhibitory activity of class 1b peptides toward two main proton-gated ion channel subtypes. Two highly homologous peptides, Hcr 1b-2 and Hcr 1b-4, similarly inhibiting rASIC1a and demonstrating an opposite effect on rASIC3, were used for in silico exploration. Theoretical models of rASIC1a complexes with Hcr 1b-2 and Hcr 1b-4 were generated, and their analyses revealed that interaction interfaces of peptides differ significantly and consist of variable residues. Since Hcr 1b-4 is the first potentiator of rASIC3 described to date, the prediction of its binding site to ASICs would to give the basis for a deeper understanding of the ASICs’ gating.

## 5. Materials and Methods

### 5.1. Isolation and Structure Determination of Sea Anemone Peptides

The specimens of *H. crispa* were collected from the South China Sea, Vietnam (2013). The species of sea anemone were identified by Dr. E. Kostina (A.V. Zhirmunsky National Scientific Center of Marine Biology FEB RAS, Vladivostok, Russia). Sea anemones were frozen and kept at –20 °C. Peptides were extracted from a whole body of sea anemone *H. crispa* and isolated as described in [25].

### 5.2. Circular Dichroism Spectra

CD spectra were measured in the range of 320 to 190 nm with a Chirascan-plus spectropolarimeter (Applied Photophysics, UK). The instrument was calibrated with 10-camphorsulfonic acid ammonium salt, and a ratio of 2:1 was found between the positive CD band at 290 nm and the negative CD band at 192 nm. The measurements were carried out in bi-distilled water at 20 °C using 0.1 and 1 cm path length cells in amide and aromatic regions, respectively. In the amide region (190–240 nm) of CD spectrum molar ellipticity, [θ]_λ_ (deg×cm^2^×dmol^−1^) was calculated as [θ]_λ_ = θ_obs_ × MRW / 10 × C × L, where θ_obs_ is the observed ellipticity in degrees; MRW is a mean amino acid residue weight of 110 Da; L is a path length (cm); C is the protein concentration (mg/mL). In the aromatic region (240–320 nm) of CD spectrum results were expressed as molar ellipticity (average molecular masses of Hcr 1b-2, Hcr 1b-3, and Hcr 1b-4 are 4522, 4523, and 4697 Da, respectively). The secondary structure contents were determined according to the method of Provencher and Glöckner, using their CONTIN program [55].

### 5.3. Electrophysiology

Animal care and animal experiments were performed following the protocol approved by the Animal Care and Use Committee of the Shemyakin-Ovchinnikov Institute of Bioorganic Chemistry, Approval Code: 267/2018; Approval Date: 2 October 2018. RAS. Rat ASIC1a and ASIC3 channels were expressed in *Xenopus laevis* oocytes after injection of 2.5–10 ng of cRNA, as previously described [56]. After injection oocytes were kept for 2–3 days at 19 °C and then up to 5 days at the temperature of 15–16 °C in sterile ND96 medium (96 mM NaCl, 2 mM KCl, 1.8 mM CaCl_2_, 1 mM MgCl_2_, 5 mM HEPES titrated to pH 7.4 with NaOH supplemented with 50 μg/mL of gentamycin). Two-electrode voltage clamp recordings were performed using GeneClamp 500 amplifier (Axon Instruments, Foster City, CA, USA). The data were filtered at 100 Hz and digitized at 1000 Hz by an AD converter L780 (LCard, Moscow, Russia) using in-house software. The solutions were applied to a cell chamber (volume 75 μL). The laminar flow of an external solution of ND96 (pH 7.8) was used at a rate of 1 mL/min. ASIC1a and ASIC3 were activated by a short pH drop from 7.8 to 5.5 or to 4.0 (10 mM MES or 10 mM acetic acid was used as the buffer solution, respectively) in supplementary 0.1% BSA solution. Peptides were applied for up to 15 s before the activation impulse. A value of currents inhibition was calculated as the ratio of the peak current amplitude when the peptide was applied to the average amplitude of the control peak currents before and after the peptide application and expressed as a percentage. To construct the concentration–response curves, a logistic equation was the following: y = ((1 – A)/(1 + ([C]/IC_50_) nH)) + A; where y is the relative value of current inhibition; C is the peptide concentration; IC_50_ is the half maximal inhibitory concentration; nH is the Hill coefficient; A is the response value at maximal inhibition (% of control). Calculations were carried out using software Origin 7.0. The data are presented as mean ± SD.

### 5.4. Homology Modeling

Theoretical models of peptide Hcr 1b-2 (UniProt accessions C0HL52) were built with homology modeling method based on 3D structures of APETx2 (PDB ID 1WXN and 2MUB) [31,38], APETx1 (1WQK) [39], and BDS-I (1BDS and 2BDS) [40]. Five models of Hcr 1b-2 were constructed and analyzed using Chimera 1.11.2rc software [57]. Models based on 1WXN spatial structure of APETx2 had minimal root mean square deviation (RMSD) values and no conformational hindrance, so this 3D structure was chosen as template for homology modeling of Hcr 1b-2 and Hcr 1b-4 (UniProt: C0HL54). Homology model of Hcr 1b-3 (UniProt: C0HL53) was generated from Hcr 1b-2 model by mutagenesis using Discovery studio 4.0 Visualizer software (BIOVIA, San Diego, CA, USA,) [41]. Analysis of the elements of secondary structure was performed with MOLMOL software (version 2k.2) [58]. The protein surface properties topographical analysis of HcrTxs 1b-2, -3, -4, and APETx2 peptides was performed with Patch analysis suite in MOE 2019.0102 CCG^®^ software (Chemical Computing Group ULC, Montreal, QC, Canada) [42].

The homology model of homotrimeric rASIC1a (construct Ser40-Leu463 aa) was generated with Modeller 9.19 interface in UCSF Chimera 1.11.2 [59], and loop refinement was performed with MOE 19.0102 CCG^®^ software [42]. The structure of chicken ASIC1a channel (cASIC1a) in a resting state at high pH (at pH 8.5–9.5) in the presence of Ca^2+^ (PDB ID 6AVE and 5WKV; resolution, 3.7 and 3.2 Å, respectively) was used as a template [15]. According to the BLAST server, the sequence identity of rASIC1a with cASIC1 was 90%. Intracellular C- and N-termini were omitted in rASIC1a theoretical model. The model’s quality was evaluated with MOE 19.0102 CCG^®^ software and the PROCHECK server [60]. The model was created at neutral pH 7.8 and optimized by molecular dynamics (MD) simulations.

### 5.5. Protein–Protein Docking Protocol

Modeling of macromolecular complexes rASIC1a–Hcr 1b-2 and rASIC1a–Hcr 1b-4 was done with the molecular docking technique. Only extracellular regions of rASIC1a trimer (Tyr71-Tyr424 aa) was used for the docking procedure. Initially blind rigid body docking was performed with the ClusPro 2.0 server [61]. The top 10 or 15 most populated clusters with the 9 Å C-alpha RMSD radius of 1000 low energy peptide positions for each dominated forces type (of 10^9^ computed docking poses) were reclustered and analyzed using Chimera 1.11.2rc [59], Discovery studio 4.0 Visualizer [41], and MOE 19.0102 CCG^®^ [42] software. Then two alternative models of rASIC1a–Hcr 1b-2 or rASIC1a–Hcr 1b-4 complexes distinct in the peptide pose were used as primary data for the ToxDock docking algorithm that was developed specially to dock peptide toxins into the potential binding site of ion channels [62]. The most energetically favorable of highly populated complexes that satisfied the constraints derived from in vitro experiments underwent MD simulations and were analyzed in more details.

### 5.6. Molecular Dynamics Simulation Protocol

MD simulations of the rASIC1a–Hcr 1b-2 and rASIC1a–Hcr 1b-4 complexes in a hydrated nonfixed 1,2-dipalmitoyl-sn-glycero-3-phosphocholine (DPPC) lipid bilayer performed in an Amber12:EHT force field [63] were carried out in accordance with the protocol of electrophysiological experiments (described in Section 5.3) at pH 7.8 and the following MD simulation at pH 5.5 under constant pressure with 300 K using the software MOE 19.0102 [42] for 300 ns for each stage. The complexes’ starting states derived from docking were embedded into a DPPC lipid bilayer of 39.9 ± 0.6 Å thickness (measured from phosphate particle to phosphate particle) constructed with a MemProtMD resource [64]. Then, the system contained 554 lipid molecules and was solvated (63,965 water molecules) and neutralized in a box 106 × 106 × 162 Å. Prior to MD simulations, the whole system was protonated (according to pH 7.8 or 5.5) with the Protonate3D tool of MOE 19.0102 software, then energy-minimized and equilibrated to reduce initial bad contacts. Equilibration consisted in the initial side chains position optimization with fixed backbone atoms, followed by a minimization with restrained carbon alpha atoms and a short molecular dynamic (100 ps). Analysis of the contact surfaces of theoretical complexes and contribution of various amino acid residues to the formation of intermolecular interface as well as visualization of the rASIC1a–Hcr 1b-2 and rASIC1a–Hcr 1b-4 complexes was performed with MOE 19.0102 software [42].

## Figures and Tables

**Figure 1 toxins-12-00266-f001:**
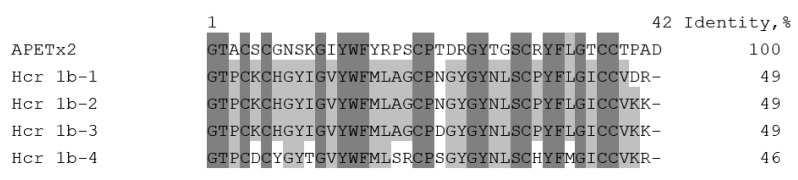
Multiple sequence alignment of the toxins: Hcr 1b-1 (P0DL87), Hcr 1b-2 (C0HL52), Hcr 1b-3 (C0HL53), Hcr 1b-4 (C0HL54) from *H. crispa*, and APETx2 (P61542) from *A. elegantissima*. Identical and conserved amino acid residues are shown on a dark and light gray background, respectively. Vector NTI software (Invitrogen, USA) [37] was used for multiple sequence alignment.

**Figure 2 toxins-12-00266-f002:**
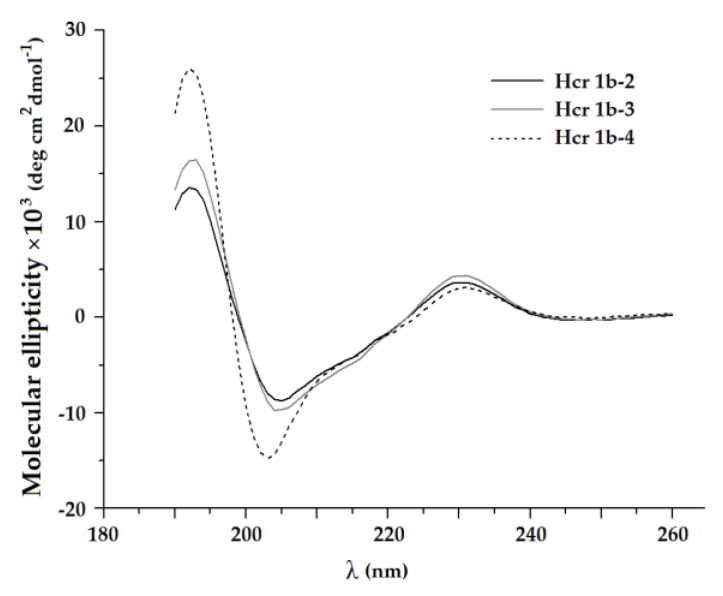
Circular dichroism (CD) spectra of *H. crispa* peptides Hcr 1b-2, -3, and -4.

**Figure 3 toxins-12-00266-f003:**
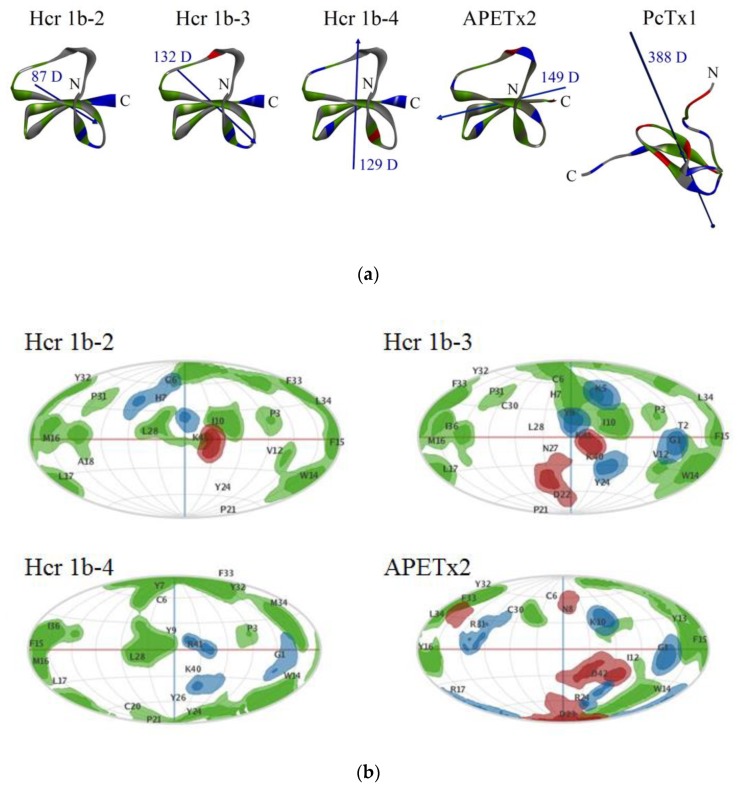
Homology models of Hcr 1b-2, -3, -4 and spatial structures of APETx2 and PcTx1. (**a**) Ribbon representation of peptide molecules and hydrophobic, basic, and acidic residues are colored green, blue, and red, respectively. Dipole moments are shown as blue arrows; magnitude of dipole moments is indicated as Debye. Visualization is performed with Discovery studio 4.0 Visualizer software [41]. (**b**) The spherical projection maps of surface electrostatic and hydrophobic properties for Hcr 1b-2, -3, -4, and APETx2 peptides performed with Patch analysis suite in MOE 2019.0102 CCG^®^ software [42]; the molecules on panel (**b**) are presented in one orientation which shows the functionally important residues of APETx2. The residue projections are labeled in a one letter code. The hydrophobic, basic, and acidic areas are presented as green, blue, and red, respectively.

**Figure 4 toxins-12-00266-f004:**
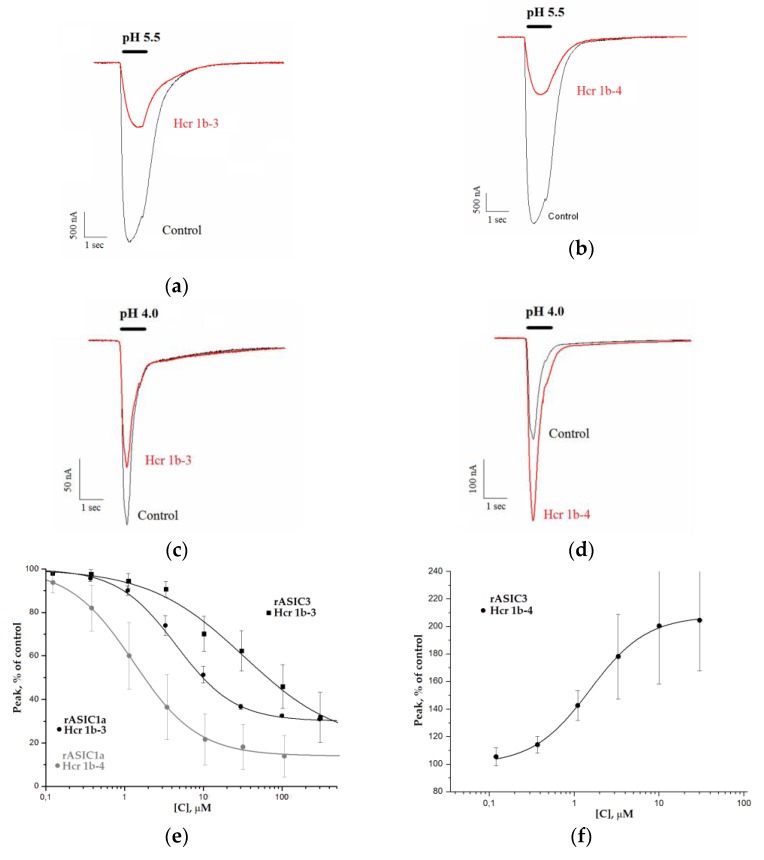
Modulatory activity of peptides Hcr 1b-3 and Hcr 1b-4 toward rASIC1a and rASIC3 channels. Acid-induced currents through rASIC1a (**a**,**b**) and rASIC3 (**c**,**d**) expressed in *X. laevis* oocytes were evoked by pH drop from 7.4 to 5.5 and 4.0; the effect of Hcr 1b-3 (**a**,**c**) and Hcr 1b-4 (**b**,**d**) at a concentration of 10 μM on the transient currents. Concentration-response curves for rASIC1a and rASIC3 indicating the inhibitory effect of peptides (**e**) and potentiating effect of Hcr 1b-4 on rASIC3 channels (**f**). Each point is means ± SD (obtained from 5 cells for each rASIC1a and rASIC3). Data were fitted by a logistic equation. The resulting values of the fitting parameters for rASIC1a: IC_50_ 4.95 ± 0.19 μM; nH of 1.20 ± 0.05; A 30.3 ± 0.7% (Hcr 1b-3) and IC_50_ 1.25 ± 0.04 μM; nH 1.08 ± 0.03; A 13.7 ± 0.8% (Hcr 1b-4). The resulting values of the fitting parameters for rASIC3: IC_50_ 17 ± 5.8 μM; nH of 0.9 ± 0.2; A 26 ± 7%; (Hcr 1b-3) and EC_50_ 1.53 ± 0.07 μM; nH 1.30 ± 0.06.; A 207.78 ± 1.56% (Hcr 1b-4).

**Figure 5 toxins-12-00266-f005:**
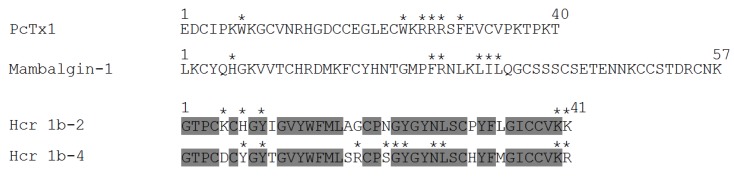
Amino acid sequences of ASIC1a and/or ASIC3 inhibitors from spider, snake, and sea anemone. Active sites are indicated by asterisks above the residues: PcTx1, mambalgin-1, and APETx2 residues that, when mutated to alanine, have a major impact on the peptides inhibitory activity toward ASICs [13,31,45,46], and Hcr 1b-2 and Hcr 1b-4 residues forming hot spot interactions. Homologous sea anemone peptides are grouped and identical amino acid residues are shown on a dark gray background.

**Figure 6 toxins-12-00266-f006:**
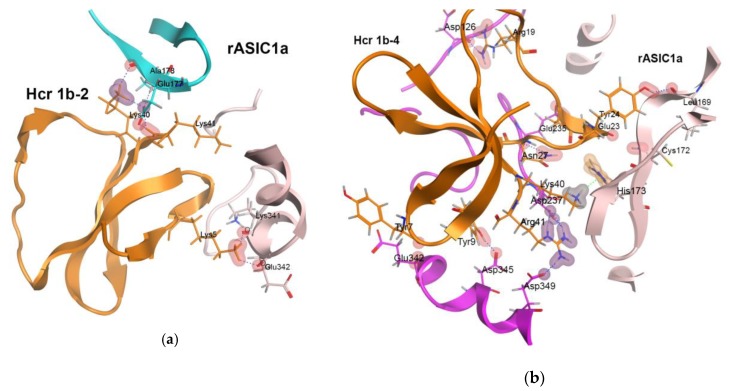
Interaction of Hcr 1b-2 and Hcr 1b-4 peptides with rASIC1a. Ribbon diagrams of direct intermolecular interactions of rASIC1a with Hcr 1b-2 (**a**) and Hcr 1b-4 (**b**,**c**) are shown. (**c**) The abnormally expanded acidic pocket of rASIC1a channel in complex with HcrTxs is shown. The distances separated Cα atoms of Asp237, Asp345, and Asp349 (proton sensors) in Hcr 1b-4–rASIC1 complex at pH 5.5 are shown. The non-covalent intermolecular interactions of Hcr 1b-4 C-terminal residues are also shown. (**d**) Network of intra- and inter-subunit ionic and hydrogen bonds between Glu277‒Arg279, Gln278‒Arg369‒Glu79, and Glu416‒Gln278 are shown. It forms a ring in the region of the central vestibule and prevents the increase in channel pore size in the complex with HcrTxs and stabilizes the closed pore state. Figure style: side chains of peptides and channel residues involved in binding are represented as sticks; hydrogen bonds as blue dotted line; ionic interactions are represented as blue colored contours as well as the hot spot interactions as magenta-colored contours; color intensity is proportional to the energy contribution.

**Table 1 toxins-12-00266-t001:** The content of the secondary structure of *H. crispa* peptides Hcr 1b-2, -3, and -4. The indices R, D, and tot refer to regular, distorted structures and their sum, respectively.

Peptide	α-Helices, %			β-Sheets, %			β-Turns, %	Unordered, %
	α_R_	α_D_	α_tot_	β_R_	β_D_	β_tot_		
Hcr 1b-2	2.3	1.8	4.1	25.5	15.9	41.4	21.5	33.0
Hcr 1b-3	1.1	2.2	3.3	26.2	16.1	42.3	23.9	30.5
Hcr 1b-4	0.2	7.7	7.9	23.0	19.7	42.7	22.4	27.0

**Table 2 toxins-12-00266-t002:** Electrophysiological activity of sea anemone toxins of structural class 1b on homomeric acid-sensing ion channels (ASICs).

Peptide	ASIC1a	ASIC3	
APETx2	-	↓ IC_50_ 0.063 μM	[32]
		↓ IC_50_ 0.175 μM *	
Hcr 1b-1	no data	↓ IC_50_ 5.5 ± 1.0 μM *	[24]
Hcr 1b-2	↓ IC_50_ 4.8 ± 0.3 μM	↓ IC_50_ 15.9 ± 1.1 μM	[25]
Hcr 1b-3	↓ IC_50_ 4.95 ± 0.19 μM	↓ IC_50_ 17 ± 5.8 μM	this work
Hcr 1b-4	↓ IC_50_ 1.25 ± 0.04 μM	↑ EC_50_ 1.53 ± 0.07 μM	this work

* IC_50_ values for APETx2 and Hcr 1b-1 were obtained toward human ASIC3 expressed in simian-kidney-derived COS-7 cells [32] and *X. laevis* oocytes [24], respectively. All other electrophysiological experiments were performed with rat ASIC1a and ASIC3 channels expressed in *X. laevis* oocytes. Inhibitory and potentiating effects are shown as ↓ and ↑, respectively.

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
