# Peer review of "APETx-Like Peptides from the Sea Anemone Heteractis crispa, Diverse in Their Effect on ASIC1a and ASIC3 Ion Channels"

_toxins, 2020, doi:10.3390/toxins12040266_

Round 1

Reviewer 1 Report

In this study, the authors showed that while the peptide toxin from sea anemone Hcr 1b-4 inhibits Asic1a, it potentiates the current of Asic3. This observation is novel and quite surprising result taking the high structural similarity of this peptide to the Hcr 1b-3 toxin that as other sea anemone toxins inhibit both channels. This study is well-executed, and the data is presented explicitly. The results support the conclusions, and the methodology is reliable. I have only one concern regarding the result interpretation regarding the toxins washout. The authors should explain in the text how the complete washout means that the toxins did not interfere with the channel conductance (line 182). This is quite an interesting conclusion that is not clear to me. How is the washout rate related to the channel conductance? I would omit this conclusion if the authors cannot provide a broad explanation of this conclusion.   

Author Response

Dear Reviewer,

We appreciate you for your special attention and kindly spirit to our manuscript.

According to your remark regarding channel conductance and reversible effect of peptide, the following statement was omitted (Please see the attachment):

Lines 181-182: This suggests that the peptide inhibits the channel without any change of its conductance.

Reviewer 2 Report

In the current manuscript, the authors report differential effects of the sea anemone peptides on ASIC1a and ASIC3 channels. The advantage of the presented work lies in the well-performed electrophysiological and extensive modeling studies to clarify the action and the binding site of H. crispa-derived peptides. However, the article misses a few critical spots that beg for answers and would make the article much more intricating. Therefore, I suggest a major revision to give the authors an opportunity to perform additional experiments.

Major comments:

  1. The authors claim that the positively charged C-termini of the peptides bind to the ASIC1a channel. If that is the case, then the binding of the peptides would be pH-dependent itself. Therefore, to reinforce their claim, it would be plausible to perform additional electrophysiological measurements to investigate the pH-dependence of channel modulation on a range of pH values. Those measurements could then be augmented with docking measurements on different pH.
  2. A quite interesting finding of the current work is the stimulatory effect of the Hcr 1b-4 peptide on ASIC3. Sadly, the effect there was entirely descriptive and lacks possible explanation regarding the binding site. To address this matter, it would be highly relevant to perform additional electrophysiological measurements to better characterize this stimulatory effect, e.g. competition experiments with other modulators with known/better characterized binding site; or checking other ASICs besides rASIC; etc... The pH dependency of activation could be considered as well and provide information whether protonation of the binding site is of any importance.
  3. A valuable missing data point in Table 2 is how Hcr 1b-1 acts on ASIC1a. As its potency would provide further insight regarding to the binding site of the peptides, this point should be added and discussed accordingly.
  4. The DEG/ENaC superfamily has other prominent members besides the ASICs, and the investigated Hcr peptides themselves have relatively high IC50 Thus, it would be relevant to discuss whether the investigated peptides are at selective for ASICs or not. If available, please provide experimental evidence regarding selectivity over other members.

Minor comment:

  • The modeling experiments on different pH values are not detailed in the Methods section. How were they performed?
  • Fig 3a: please consider color-coding the parts of the ribbons for better representing the electrostatics using the color code in 3b.
  • Please revise the following lines for formal corrections: lines 26 (ions), 70 (in current),83-84 (italic), 127 (peptides), 136 (differese) and S1a

Author Response

Dear Reviewer,

We appreciate you for extensive revision of our manuscript and useful suggestions. We absolutely agree with the remarks that it would be reasonable to perform additional electrophysiological investigation at various pH values, deep studying of unexpected potentiating effect of Hcr 1b-4 and pharmacological profile of HcrTxs. We are very interested in conduction of this study since it undoubtedly contributes substantially to the understanding of the HcrTxs action mechanism. The main experiments concerning with your important comments we plan to include in the future manuscript focused on molecular design of ligands (recombinant peptides and mutant analogs, which will be obtained) and studying of their activity. The main task of this work was to state the unusual effect of Hcr 1b-4, as a natural molecule, among APETx-like peptides as well as to explain and compare the inhibitory effect of HcrTxs on rASIC1a.

Please find below our answers on your comments. Sincerely hope that we could clarify some points, but generally, due to the current COVID-19 epidemiological situation, we are under quarantine and unable to conduct electrophysiological experiments in the next two months. In addition, it is not clear how soon after quarantine ends we will be able to restart experimental work.

We thank you again for your special attention and your time.

Major comments:

  1. The authors claim that the positively charged C-termini of the peptides bind to the ASIC1a channel. If that is the case, then the binding of the peptides would be pH-dependent itself. Therefore, to reinforce their claim, it would be plausible to perform additional electrophysiological measurements to investigate the pH-dependence of channel modulation on a range of pH values. Those measurements could then be augmented with docking measurements on different pH.

Previously, the crucial contribution of positively charged amino acid residues of ASICs modulators was postulated in a number of publications [1-3] and ascertained thanks to determination of the toxin-channel complex spatial structures and electrophysiological investigation of mutant peptides [4-9]. Analysis of HcrTxs structures and molecular modeling of their complexes with the rASIC1a channel performed by us show complete agreement with this statement. For several ASICs ligands, data, demonstrating the influence of different pH values on channel modulation, was reported to clarify the mechanism of ASICs activation [1].

In this manuscript, we would not like to lay emphasis on the ion channel functioning, but characterize and compare the potency and efficacy of peptides Hcr 1b-2 – Hcr 1b-4 as well as focus on their intriguing interaction with ASIC1a subtype. However, in continuation of this work we assume to conduct such experiments. When novel data of electrophysiological experiments showing influence of pH values on HcrTxs modulatory effects will be obtained molecular modeling would significantly promote determination of HcrTxs binding mode of peptides in view of lack of experimental data on 3D-structure of rASIC1a–Hcr 1b-2 and rASIC1a–Hcr 1b-4 complexes.

  1. Baron, A.; Diochot, S.; Salinas, M.; Deval, E.; Noël, J.; Lingueglia, E. Venom toxins in the exploration of molecular, physiological and pathophysiological functions of acid-sensing ion channels. Toxicon 2013, 75, 187–204.
  2. Báez, A.; Salceda, E.; Fló, M.; Graña, M.; Fernández, C.; Vega, R.; Soto, E. α-Dendrotoxin inhibits the ASIC current in dorsal root ganglion neurons from rat. Neurosci. Lett. 2015, 606, 42–47.
  3. Vullo, S.; Kellenberger, S. A molecular view of the function and pharmacology of acid-sensing ion channels. Pharmacol. Res. 2019, 154, 104166.
  4. Baconguis, I.; Gouaux, E. Structural plasticity and dynamic selectivity of acid-sensing ion channel–spider toxin complexes. Nature 2012, 489, 400–405.
  5. Dawson, R.J.P.; Benz, J.; Stohler, P.; Tetaz, T.; Joseph, C.; Huber, S.; Schmid, G.; Hügin, D.; Pflimlin, P.; Trube, G.; et al. Structure of the Acid-sensing ion channel 1 in complex with the gating modifier Psalmotoxin 1. Nat. Commun. 2012, 3, 936.
  6. Sun, D.; Yu, Y.; Xue, X.; Pan, M.; Wen, M.; Li, S.; Qu, Q.; Li, X.; Zhang, L.; Li, X.; et al. Cryo-EM structure of the ASIC1a-mambalgin-1 complex reveals that the peptide toxin mambalgin-1 inhibits acid-sensing ion channels through an unusual allosteric effect. Cell Discov. 2018, 4, 27.
  7. Jensen, J.E.; Cristofori-Armstrong, B.; Anangi, R.; Rosengren, K.J.; Lau, C.H.Y.; Mobli, M.; Brust, A.; Alewood, P.F.; King, G.F.; Rash, L.D. Understanding the Molecular Basis of Toxin Promiscuity: The Analgesic Sea Anemone Peptide APETx2 Interacts with Acid-Sensing Ion Channel 3 and hERG Channels via Overlapping Pharmacophores. J. Med. Chem. 2014, 57, 9195–9203.
  8. Saez, N.J.; Deplazes, E.; Cristofori-Armstrong, B.; Chassagnon, I.R.; Lin, X.; Mobli, M.; Mark, A.E.; Rash, L.D.; King, G.F. Molecular dynamics and functional studies define a hot spot of crystal contacts essential for PcTx1 inhibition of acid-sensing ion channel 1a. Br. J. Pharmacol. 2015, 172, 4985–4995.
  9. Mourier, G.; Salinas, M.; Kessler, P.; Stura, E.A.; Leblanc, M.; Tepshi, L.; Besson, T.; Diochot, S.; Baron, A.; Douguet, D.; et al. Mambalgin-1 Pain-relieving Peptide, Stepwise Solid-phase Synthesis, Crystal Structure, and Functional Domain for Acid-sensing Ion Channel 1a Inhibition. J. Biol. Chem. 2016, 291, 2616–2629.

  1. A quite interesting finding of the current work is the stimulatory effect of the Hcr 1b-4 peptide on ASIC3. Sadly, the effect there was entirely descriptive and lacks possible explanation regarding the binding site. To address this matter, it would be highly relevant to perform additional electrophysiological measurements to better characterize this stimulatory effect, e.g. competition experiments with other modulators with known/better characterized binding site; or checking other ASICs besides rASIC; etc... The pH dependency of activation could be considered as well and provide information whether protonation of the binding site is of any importance.

We are grateful for the offers designed to further insight into Hcr 1b-4 potentiating activity, and we are glad to know that unusual properties of this peptide were positively evaluated. Electrophysiological investigation of Hcr 1b-4 and other HcrTxs reported in the manuscript were carried out using native peptides, which amount, unfortunately, have been limited. Continuation of the experiments requires fairly large amount of peptide, and, therefore, it is necessary to obtain a recombinant peptide for further investigation. Currently we are working on developing a scheme of recombinant peptide production as well as mutant Hcr 1b-2, in which some residues, proposed to be functionally important, are replaced to residues of Hcr 1b-4 to obtain chimeric peptides with desirable activity. We are going to perform competition experiments using wild-type and mutant peptides as well as study of HcrTxs effects on human ASIC3 to better understand peptide interaction with ASIC3 channels. At the same time we believe that the reliable evidence of HcrTxs binding site localization may be derived from crystallographic analysis or cryo-electron microscopy, which is rapidly progressing and has already contributed to the determination of the mambalgin-1 binding site with cASIC1.

  1. A valuable missing data point in Table 2 is how Hcr 1b-1 acts on ASIC1a. As its potency would provide further insight regarding to the binding site of the peptides, this point should be added and discussed accordingly.

In view of contradictive effect of H. crispa peptides, Hcr 1b-2 – Hcr 1b-4, and toxin APETx2 on homomeric ASIC1a channels, we have made attempt to test if Hcr 1b-1 affects ASIC1a. Electrophysiological experiments, similar to those described in the manuscript, were conducted, using native Hcr 1b-1. However, unambiguous results, allowing to declare any effect of Hcr 1b-1 on ASIC1a subtype or absence of activity, were not obtained. We believe that these confusing results occurred due to unstable ion channel expression. So, current data, regarding activity of APETx-like peptides on ASIC1a, were given in table 2. While toxin APETx2 was shown to be inactive on ASIC1a, there is no reliable data about activity of Hcr 1b-1 toward ASIC1a channel. After the end of the quarantine measures designed to prevent the spread of COVID-19 we are going to return to work on recombinant Hcr 1b-1 production for continuation of the electrophysiological experiments in order to solve the question of Hcr 1b-1 effect on ASIC1a.

  1. The DEG/ENaC superfamily has other prominent members besides the ASICs, and the investigated Hcr peptides themselves have relatively high IC50. Thus, it would be relevant to discuss whether the investigated peptides are at selective for ASICs or not. If available, please provide experimental evidence regarding selectivity over other members.

We are planning to finish testing of HcrTxs effect toward a number of voltage-gated cation channels, targeted by the other APETx-like peptides, APETx1 – APETX4 and BDS toxins. We believe that the relatively high values of IC50 reported in this manuscript indicate that HcrTxs may be more effective on the other molecular target(s).

We would like to express our gratitude for the idea to test HcrTxs against the other members of DEG/ENaC superfamily. It is intriguing problem, especially in the light of absence of any results regarding activity of APETx-like peptides on any DEG/ENaC ion channels, except for ASICs, as far as we know.

Taking into account the comment we discussed in the manuscript the possibility of the existence of undescribed HcrTxs molecular target(s):

Lines 334-350: “As we previously mentioned, HcrTxs are weak ASICs modulators demonstrating their effects at micromolar concentration. At the same time APETx-like peptides were shown to modulate several ion channels, ASICs, NaV, and KV. Toxin APETx2 inhibits ASIC3 (IC50 63 nm [32]) as well as voltage-gated sodium channels NaV1.2 (IC50 114 ± 25 nm) and NaV1.8 (IC50 55 ± 10 nM) [30] at nanomolar concentrations, but higher concentrations of APETx2 inhibit voltage-gated potassium channel hERG (IC50 1.21 ± 0.05 μM) [31]. Unfortunately, it was not reported if other APETx-like peptides affect ASICs. Highly homologous peptides APETx1, APETx3, and APETx4 primarily inhibit Kv11 (IC50 34 nm for hERG [33]), potentiate TTX-sensitive Nav (EC50 2265 ± 185 nm for NaV1.6 [30]), and inhibit KV10.1 (IC50 1.1 μM [29]), respectively, but, in general, each of these peptides modulate several subtypes of NaV and KV channels. Finally, toxin BDS-I initially characterized as KV3 channel blocker (IC50 values 47 nM [51] and 220nm [52] for KV3.1 and KV3.4, respectively) also potentiates TTX-sensitive NaV channels (EC50 about 3 nM for NaV1.7) and weakly inhibits TTX-resistant NaV [53]. Collectively, these data evidence that HcrTxs may modulate ion channels other than ASICs at lower concentration range. It is noteworthy that ion channels targeted by APETx-like peptides belong to two different superfamilies: voltage-gated ion channels (NaV and KV) and DEG/ENaC (ASICs). As far as we knows, there is no data describing modulation of any member of DEG/ENaC superfamily, except for ASICs, by APETx-like peptides.”

Due to the changes made, three references were added to the reference list under the numbers 51-53; consequently, the links numbered 51-60 are currently numbering 54-63. As we used Mendeley software, changes in references, unfortunately, can’t be shown in the marked version.

  1. Diochot, S.; Schweitz, H.; Béress, L.; Lazdunski, M. Sea anemone peptides with a specific blocking activity against the fast inactivating potassium channel Kv3.4. J. Biol. Chem. 1998, 273, 6744–6749, doi: 10.1074/jbc.273.12.6744.
  2. Yeung, S.Y.M. Modulation of Kv3 subfamily potassium currents by the sea anemone toxin BDS: significance for CNS and biophysical studies. J. Neurosci. 2005, 25, 8735–8745, doi: 10.1523/JNEUROSCI.2119-05.2005.
  3. Liu, P.; Jo, S.; Bean, B.P. Modulation of neuronal sodium channels by the sea anemone peptide BDS-I. J. Neurophysiol. 2012, 107, 3155–67, doi: 10.1152/jn.00785.2011.

Minor comments:

  1. The modeling experiments on different pH values are not detailed in the Methods section. How were they performed?

In electrophysiological experiments HcrTxs were applied at a pH of 7.8, when the channel is in a closed state. Then the channel was activated by dropping the pH to 5.5 value. In our modeling experiments, we tried to reconstruct these conditions and simulated MD in two stages, first with a protonated system at pH 7.8, and then continued after the system was reprotonated at pH 5.5. In accordance with your recommendations, we have corrected the Molecular dynamics simulation protocol section.

Lines 500-508: “MD simulations of the rASIC1a–Hcr 1b-2 and rASIC1a–Hcr 1b-4 complexes in a hydrated POPC lipid bilayer performed in an Amber12:EHT force field [60] were performed under conditions of constant pressure, 300 K, and pH 7.8 and 5.5 using MOE 19.0102 software [42] for 100 ns. Prior to MD simulations, whole system was energy minimized and equilibrated to reduce initial bad contacts.”was corrected to

“MD simulations of the rASIC1a–Hcr 1b-2 and rASIC1a–Hcr 1b-4 complexes in a hydrated POPC lipid bilayer performed in an Amber12:EHT force field [63], were carried out in accordance with the protocol of electrophysiological experiments (described in section 5.3) at pH 7.8 and the following MD simulation at pH 5.5 under constant pressure, 300 K using the software MOE 19.0102 [42] for 300 ns for each stage. Prior to MD simulations, whole system was protonated (according to pH 7.8 or 5.5) with Protonate3D tool of MOE 19.0102 software, then energy minimized and equilibrated to reduce initial bad contacts.”

  1. Fig. 3a: Please consider color-coding the parts of the ribbons for better representing the electrostatics using the color code in 3b.

The figure 3a was corrected (Please see the attachment) and following comment was added.

Lines 152-153: Ribbon representation of peptide molecules, hydrophobic, basic, and acidic residues are colored green, blue, and red, respectively.

  1. Please revise the following lines for formal corrections: lines 26 (ions), 70 (in current), 83-84 (italic), 127 (peptides), 136 (differese) and S1a.

The following changes were made in accordance with given remarks:

Line 26: “Acid-sensing ions channels (ASICs) activated by the extracellular pH decrease” was corrected to “Acid-sensing ion channels (ASICs) activated by the extracellular pH decrease”.

Line 70: “In current work” was corrected to “In this work”.

Lines 78-84: The specific names of sea anemones were print in italic “Among the great diversity of sea anemone peptide structures, the H. crispa toxins (HcrTxs) affecting ASIC channels should be attributed to the structural class 1b [26,27]. HcrTxs share the best identity with the sea anemone toxins of class 1b, such as antimicrobial peptide crassicorin-I from Urticina crassicornis [28], ion channel modulators APETx1 – APETx4 from A. elegantissima affecting ASICs and/or voltage-gated potassium and/or sodium channels [29–33], and several functionally uncharacterized APETx-like peptides identified in transcriptomes of sea anemones A. elegantissima, Bunodosoma granuliferum [34], Anemonia sulcata [35], and in the proteome of Bunodosoma cangicum [36].”

Line 127: “Hcr 1b-2, -3, -4, and APETx2 are very basic peptide” was corrected to “Hcr 1b-2, -3, -4, and APETx2 are very basic peptides”.

Line 135: “one can see differese in their molecule surface electrostatic and hydrophobic properties distribution” was corrected to “one can see difference in their molecule surface electrostatic and hydrophobic properties distribution.”

Figure S1 was corrected (Please see the attachment).

“Aidick pocket” was changed to “Acidic pocket”.

  1. Figures 4c and 4d (Please see the attachment) as well as lines 184-186 were corrected:

“Acid-induced currents through rASIC1a (a, b) and rASIC3 (c, d) expressed in X. laevis oocytes were evoked by pH drop from 7.4 to 5.5; the effect of Hcr 1b-3 (a, c) and Hcr 1b-4 (b, d) at a concentration of 10 μM on the transient currents” was corrected to

“Acid-induced currents through rASIC1a (a, b) and rASIC3 (c, d) expressed in X. laevis oocytes were evoked by pH drop from 7.4 to 5.5 and 4.0; the effect of Hcr 1b-3 (a, c) and Hcr 1b-4 (b, d) at a concentration of 10 μM on the transient currents.”

Round 2

Reviewer 2 Report

The authors answered clearly to the questions that have arisen during the revision process, and discussed them extensively. Moreover, they have re-worked the few unclear points within the figures. Due to the recent pandemic situation it would be indeed unfair to wait for further electrophysiological clarification - so I propose that those can also be incorporated into a follow-up of this work. 

In the light of the statements above, I suggest to accept the current manuscript in this revised form after minor changes.

Minor comments: The authors included in the revised manuscript that the MD simulations were performed in a POPC bilayer. In this case it would be useful to add further methodological information, namely bilayer thickness, whether it was surrounded by an aqueous solution or not, what the dimensions of the periodic cell were, whether the bilayer was fixed or not during the MD simulation. 

As the data is available, one thing could enhance the readability for non-ASIC experts - illustrating the putative binding of the peptide to the membrane-incorporated channel model (cross-section?) as an additional panel of Figure S1.

Spelling errors: line 348 ... as far as we know...; MD method: POPC abbreviation missing

Author Response

Dear Reviewer,

We are truly grateful for your special attention and kindly spirit to our manuscript.

According to your remarks, the following changes were made:

  1. Taking into account your comment, molecular dynamics simulation protocol section has been corrected:

Line 498: “POPC lipid bilayer” was corrected tononfixed 1,2-dipalmitoyl-sn-glycero-3-phosphocholine (DPPC)”, and following information was inserted:

Lines 502-505: “The complexes starting states derived from docking were embedded into a DPPC lipid bilayer of 39.9 ± 0.6 Å thickness (measured from phosphate particle to phosphate particle) constructed with MemProtMD resource [64]. Then, the system contained 554 lipid molecules was solvated (63965 water molecules) and neutralized in a box 106×106×162 Å.”

Due to the changes made, reference [64] was added:

Lines 709-711: 64. Newport, T.D.; Sansom, M.S.P.; Stansfeld, P.J. The MemProtMD database: a resource for membrane-embedded protein structures and their lipid interactions. Nucleic Acids Res. 2019, 47, D390–D397. doi: 10.1093/nar/gky1047.

  1. The figure S1b (Please see the attachment) and following comments were added:

Line 518: “3D-structure models of Hcr 1b-2 (a) and Hcr 1b-4 (b) complexes with rASIC1a. (a)”

Lines 523-530: “Visualization is performed with Discovery studio 4.0 Visualizer software [41].” was changed to “(b) Ribbon diagram of the Hcr 1b-4–rASIC1 complex embedded into DPPC lipid bilayer (represented as gray sticks). The Arg19, Lys40, and Arg41 side chains of Hcr 1b-4 are represented as bold sticks, and the channel residues involved in a circle pattern of inter- and intrasubunit hydrogen bonds and ionic interactions within the ASIC1a central vestibule as well as pore residues Asp432 are represented as balls and labeled. Part of the DPPC lipid bilayer as well as solvent molecules have been removed for clarity. Visualization is performed with Discovery studio 4.0 Visualizer software [41] (a) and MOE 19.0102 [42] (b).”

  1. Line 348: “As far as we knows,” was corrected to “As far as we know,”.
  2. In addition prefix “r” was added to ASIC1a or ASIC3 abbreviation if necessary:

Lines 269, 358, 369, 372, 386, 387, 392, 398, 399, 406, 409: “ASIC1a” was corrected to “rASIC1a”.

Lines 312, 316: “ASIC3” was corrected to “rASIC3”.

Lines 374, 376: “ASIC1” was corrected to “rASIC1a”.

Lines 393, 395: “ASIC1a–Hcr 1b-2 and ASIC1a–Hcr 1b-4 complexeswas corrected to rASIC1a–Hcr 1b-2 and rASIC1a–Hcr 1b-4 complexes”.

Line 405: “desensitized cASIC1a channel” was corrected to “desensitized cASIC1 channel”.